# The development of global creative centers from a regional perspective: A case study of Milan's creative industries

**Cheng Shen[1], Xinyi Zhang[1], Xiang Li[2]***

**1** Shanghai Tongji Urban Planning and Design Institute, Shanghai, China, **2** School of Design and Architecture, Zhejiang University of Technology, Hangzhou, China

* lixiang@zjut.edu.cn

## Abstract

With the advancement of globalization and the deepening interlock between cities globally, the research on urban culture and creative development should be expanded from a single city scale to regional and urban agglomeration scale. Creative clusters will become an important spatial scale to break the bottleneck of urban development in the future. In Europe, cities such as London, Paris, and Milan have developed creative industries and built creative clusters relatively early, and hence they gradually become global creative centers. However, there are few studies on the development of their creative industries from a regional perspective. This research uses multi-source data within the scope selected and based on comprehensive standards, to analyze regional development characteristics of creative industries in Milan, creative clusters formed with Milan as its core, and influencing factors of its development. Through the study of a global creative center from a regional perspective, references for emerging creative cities and those that require industrial transformation can be achieved.

## 1. Introduction

Creative innovation that is driven by creative industries is one of the most important ways to achieve development in the post-industrial age for European cities such as London, Paris, and Milan [1]. Through industry transformation, these cities have realized the rise of creative industries, promoting the urban renewal process and creating a new driving force for city development. Along with the rapid expansion of creative industries, there have also formed far-reaching influence on regional and even global scales, so that they are gradually becoming global centers of creativity and design [2].

With the innovation clusters becoming the basic unit to achieve innovation, address development difficulties, and overcome creative bottleneck for individual cities in the new era, the regional perspective has become an important spatial scale to analyze city development [3]. The region (regional) mentioned here and in later parts of this study refers to a broad concept of a large area with unclear boundaries, but not a specific spatial unit in the European context. However, through literature reviews, it can be found that there are few quantitative studies on

**Data Availability Statement:** Data are available via the following link: https://github.com/ScaithyShen/Creative-industries-of-Milan.

**Funding:** Humanities and Social Science Promotion Plan of Zhejiang University of

Technology Pre-research Fund, Grant/Award Number: GZ2298118004, awarded to XL.

**Competing interests:** The authors have declared that no competing interests exist.

the development of global creative centers' creative industries like Milan from a regional perspective, and the scope of these issues has not been extended to as the Europe-wide scale yet.

The study of global creative centers from a regional perspective will reveal its development characteristics and influencing factors, and thus references for emerging creative cities and those that require industrial transformation can be achieved.

## 2. Literature review

### 2.1. Research on creative industries and innovation must be extended

The academic circles put forward the concept of the "cultural industry" in the 1940s, and the concept of "creative industries" in the 1990s, proposed by Department of Culture, Media, and Sport (DCMS) of the British government, and then spread it to the world. Thereafter, there was a global wave of research on creative industries, which has gradually matured since 2006 [4], basically forming three research fields: the creative cluster, class, and city. The main content can be concluded as the concept, constitution, formation, development mechanism, and factors influencing development and their impact on urban economics and society. In addition, the humanities and economics, geography, regional development, and other fields have also received extensive attention [5]. The "creative class" theory proposed [6, 7] by Richard Florida, a famous American urban economist, has been the focus of academic debate and urban policy formation [8]. Later, with the development of data and technical means, some scholars conducted empirical studies on spatial distribution characteristics of creative enterprises in different cities, and put forward convincing evidence to prove that creative industries' enterprises have specific tendencies, such as spatial agglomeration [5, 9–14].

Milan and Paris, as the world's leading creative cities, are the focus of academic circles. However, through the literature review, we found that its current research is still limited. Firstly, most of this research is focused on single cities, and there are few systematic and large-scale regional studies. Secondly, the research content is mainly macro and qualitative analyses, and there are less quantitative analyses based on data, especially from a regional perspective [15–17].

### 2.2. Urban agglomerations and regions from the perspective of space of flow

In 1989, Castells proposed the concept of "space of flow," a spatial form composed of circulation elements [18]. The "flow" as the real relational data reflects the interaction between cities and reflects the structure, function, and interlock of the urban network [19]. Sassen and Tylor believe that space of flow will replace traditional space as the future trend [20, 21]. In 2010, Taylor proposed the central flow theory, which holds that when both space of place and space of flow exist in a region, it absorbs the core viewpoint of space of flow; however, it does not completely deny the central place theory, which is explained from the perspective of functional and interlock perspectives [22]. Thereafter, Hall and some scholars hold that the characteristics of flows began to appear in Western Europe [23]. Then with the flow of people, things, information, capital, and technology between cities being accelerating [24], the research based on the concept of "space of flow" is an integral part for the analysis of regions and urban agglomerations, which has also become a hot topic and has been evolving in recent years. This prompts the research of regional spatial structure to shift from the internal characteristics of the city to the external relations of the city, and the focus to the urban network and interlock relations and studies based on various flows originated under this concept.

In terms of classification, there are real-measured flows, such as population flow and postal logistics, and parametric substitution flows, such as economic and information flows [25]. Research in this area diversifies gradually as the data improve. Early studies like that of traffic flow emerged, based on data of air or railway transportation [26], as well as enterprise flow based on data of headquarters and branches. Later, research on information flow based on bank data [27] and innovation flow based on patent data also emerged [28]. Hence, the types of data are becoming more abstract and diverse.

With the further development of technology and big data in city flow research, studies have been conducted based on data from mobile phones, internet and social media platforms [29], and mobile positioning [30], which further expanded on the research content of flow and made it possible to extend research into cyberspace [31]. In recent years, research tends to be further subdivided. For example, research based on enterprise flow began to focus on some specific types of enterprises. In summary, the evolution of data types helps strengthening the research depth. As research is gradually becoming more in-depth and diversified, however, there are few studies that focus on "enterprise flow" driven by creative industries of the world's creative centers, such as Milan and Paris.

## 3. Materials and methods

### 3.1. Research goal

The goal of this study is to analyze the development of creative industries in Milan, a global creative center, by using multi-source data within the scope of a Europe-wide region. The study takes the data of creative industry enterprises as relational data, analyzes the interlock relationship of the creative industry built by Milan and cities in the region, the characteristics and development trends of the interlock network with Milan as the core and combined with a number of indicators, to further explore the influencing factors of regional development of Milan's creative industry. Through the study of a global creative center from a regional per-spective, references for emerging creative cities and those that require industrial transforma-tion can be achieved.

### 3.2. Research scope

The case study of this study is Milan, one of the creative core cities of Europe and the scale of Milan is based on the administrative boundary determined by Milano Geoportale [32]. In order to cover the whole Europe, make the selected cities representative, and facilitate data acquisition and analysis, taking comprehensive factors such as innovation capacity, economy, urban size, population, location, and so on into consideration and referring some lists of Euro-pean cities such as the European Creative City Ranking [33], List of European Cities and Met-ropolitan Areas based on Population [34], Ranking of European Metropolitan Areas based on GDP [35] etc., we selected 52 European cities as the scope of this research. To compare the development of Milan's creative industries on a global scale, eight global cities were selected as a control group, based on the classification of GaWC City Rankings [36],which are shown in Table 1. The location in the table refers to The World Factbook [37], The division of location here is only used for the statistical convenience and does not imply any assumption regarding political or other affiliation of countries or territories.

### 3.3. Core concept definitions

Creative industries are the main industry category of this study. In a European context, the DCMS of the UK identified nine creative industry categories with wide influence and high

**Table 1. City list of the research scope.**

| Location | Name of Cities |
|---|---|
| **Southern Europe** | Milan (Italy); Rome (Italy); Madrid (Spain); Barcelona (Spain); Bilbao (Spain); Seville (Spain); Lisbon (Portugal); Porto (Portugal); Athens (Greek) |
| **Western Europe** | Pairs (France); Lyon (France); Nice (France); Marseille (France); Lille (France); Brussels (Belgium); Antwerp (Belgium); Amsterdam (Netherlands); Rotterdam (Netherlands); The Hague (Netherlands); Eindhoven (Netherlands); London (UK); Liverpool (UK); Luxembourg (Luxembourg) |
| **Central Europe** | Berlin (Germany); Munich (Germany); Stuttgart (Germany); Bremen (Germany); Hamburg (Germany); Frankfurt am main (Germany); Dusseldorf (Germany); Koln (Germany); Nuremberg (Germany); Vienna (Austria); Praha (Czech Republic); Zurich (Switzerland); Geneva (Switzerland); Basel (Switzerland) |
| **Eastern Europe** | Moscow (Russia); Saint Petersburg (Russia); Istanbul (Turkey); Warsaw (Poland); Kiev (Ukraine); Budapest (Hungary); Tallinn (Estonia); Zagreb (Croatia); Riga (Latvia); Minsk (Belarus); Belgrade (Serbia) |
| **Northern Europe** | Copenhagen (Denmark); Oslo (Norway); Helsinki (Finland); Stockholm (Sweden) |
| **World Cities (Control Group)** | New York (USA); Tokyo (Japan); Singapore (Singapore); Beijing (China); Shanghai (China); Hong Kong (China); Sydney (Australia); Dubai (UAE) |

academic recognition [38]: advertising and marketing; architecture; crafts; products, graphic, and fashion design; film, television, video, radio, and photography; software, games, and computer services; publishing; museums, galleries, and libraries; and music, performing, and visual arts. Although academic research or classification standards are slightly different for the specific names of these categories, their content are included in the nine categories determined by DCMS. Therefore, in the research we adopt these nine categories as the basis for classification and data acquisition of creative industries in Milan.

This study will analyze the evolution of regional development of Milan's creative industries based on time dimension; the choice of time span will be demonstrated. In Europe, creative industries such as fashion design first appeared in UK and France, such as the brands Burberry (UK, founded in 1856) and Hermes (France, founded in 1837). Thereafter, Italian creative industries such as PRADA (1913) and GUCCI (1921) have emerged. Studies show that from their birth to the 1960s, they resided mainly in the domestic market and did not expand to other European countries; the 1960s and 1970s were a critical turning point in the development of Italian fashion and creative industries [39]. At this time, creative industries in Milan gradually expanded from the original "quadrilatero della moda" to the entire city. The second group of the now-famous Italian fashion and design brands then began to emerge, such as VALENTINO (born in 1960), VERSACE (born in 1978), ARMANI (born in 1975), and so on. This period was also accompanied by the regional expansion of Italian creative industries [40, 41]. Therefore, the time span from 1970 to 2020 is chosen for this study.

This study will explore the factors influencing the regional development of creative industries in Milan. The "3T" theory proposed by Florida is currently recognized as the classical theory of influencing factors of creative industries [7], which includes three dimensions: talent, technology, and tolerance. Thereafter, scholars expanded on this with factors such as policies and infrastructure [42–45]. Combined with accessibility of the data, this study chooses four categories of indicators to conduct the analysis: talent, technology, tolerance, and infrastructure.

## 3.4. Data acquisition

The Data acquisition process is shown in Fig 1. The first part of the data is enterprise data of headquarters and branches. In this study, the bottom-up data acquisition method was adopted

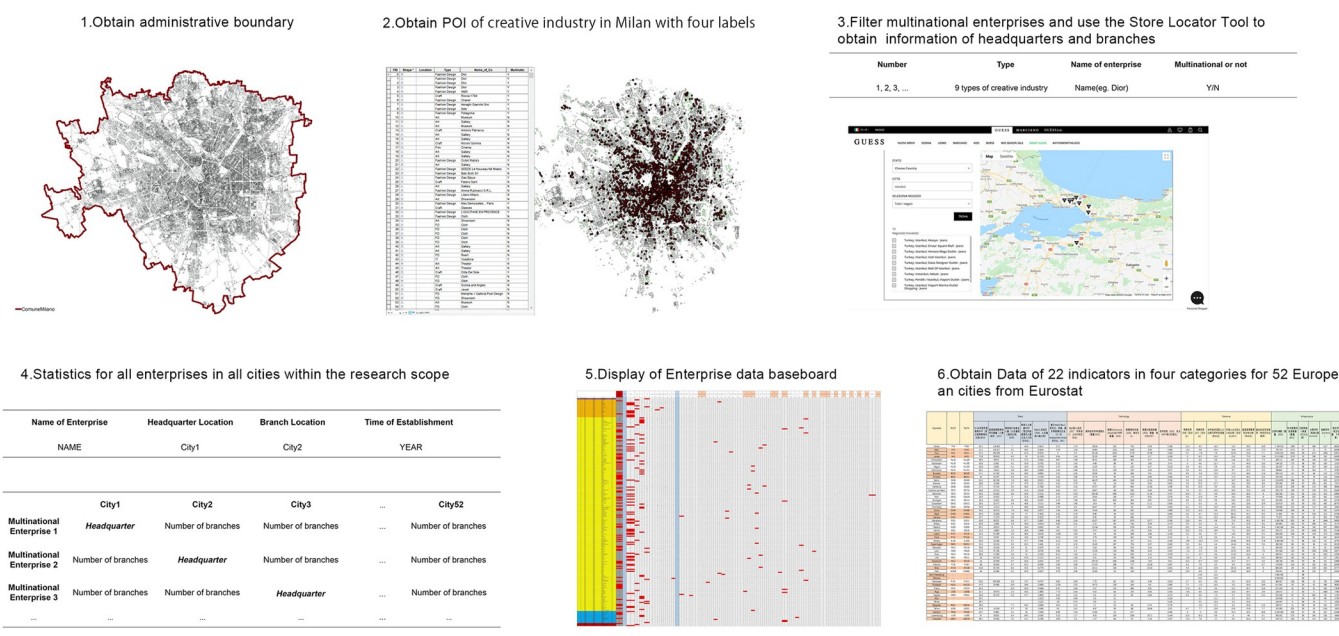

**Fig 1. Data acquisition process.**

from Milan. Firstly, basic data such as the administrative scope of Milan are obtained through Milano Geoportale [32]. Secondly, within this scope, POI data of Milan's creative industries enterprises are manually obtained through Google Map Platform [46], with three data labels that are identified and retained through screening, integration, and manual classification as follows: type (9 categories), enterprise name, and whether they are multinational (Y/N). Therefore, a total of 6,523 creative industries' POI data were collected; thereafter, after removing the repeated data of multinational enterprises, the position of their headquarters and the number of their branches in the remaining 51 cities within the scope were collected according to the positioning tool (often called the Store Locator Tool) of each enterprise's official website. A total of 22,322 valid data pieces were collected in this part. The location of the headquarters and branch are collected for each data label, such as the name of the enterprise. In addition, time dimension information can also be obtained by screening the establishment time of the enterprises.

The second part of the data is the indicator system, which is shown in Table 2: there are 20 indicators of talent, technology, tolerance, and infrastructure for the 52 cities within the research scope. They are obtained through Eurostat [47]. Eurostat uses the Nomenclature of Regional Statistical Units (NUTS), which lists 281 NUT2 and 1348 NUT3 units in NUTS 2016, as standard statistical units, or direct statistics for individual cities. Due to the availability of the data, this study is simplified and the NUTS unit data of the city was used for some indicators replacing the city data [48]. These 20 indicators also contain time span, which can be obtained from the available time range. The data of the straight-line distance between Milan and 52 cities are also obtained by the distance tool of Google Map.

### 3.5. Research design

Based on the enterprise interlocking network research method from the perspective of "flow space," taking city i as the research object, the outward interlocking value of creative industries

**Table 2. Four categories of indicators for 52 cities obtained by Eurostat.**

| Category | Indicators |
|---|---|
| Talent | Educational level of residents aged 25–64 (percentage of people with higher education, ISCED international Education Standard Classification 5–8, 2000–2019); Number of students in higher education (Number, 2010–2017); High-tech employment (as a percentage of total employment, 2010–2019); Human resources in science HRST (percentage of the population educated in science and technology, 2000–2019); R&D personnel status (as a percentage of total employment, 2000–2019); Employment in culture, creative, entertainment and other high-growth businesses (as a percentage of total employment, 2017) |
| Technology | R&D investment (percentage of GDP, 2010–2018); European Community Design Applications (Number, 2010–2016); High-tech patents (Number, 1990–2010); Eu Trademark registrations (Number/billion GDP, 2000–2010); Revenue from patents (nominal GDP per billion euros, 2010) |
| Tolerance | Foreign-born population (percentage, 2010–2017); Gender employment gap (percentage difference with men, 2010–2019); Proportion of women in higher education (percentage, 2010–2018); Percentage of economically active women (percentage difference with men, 2010–2018) |
| Infrastructure | Percentage of households with broadband access (percentage, 2010–2019); Highway construction (km/km$^2$, 2010–2017); Aviation (1,000 passengers, 2010–2019); Railway Construction (km/km$^2$, 2010–2018); Number of facilities for visitors (Quantity, 2000–2010) |

between cities i and j is as follows:

$$C(i,j) = K_0 \sum_{m=1}^{n_1} K_1 Y_m \tag{1}$$

where K0 = 2 is the weight of headquarters, K1 = 1 is the weight of branches, Y is the number of branches of an enterprise in city j whose headquarters are in city i, and n1 is the total number of enterprises with branches in city j. Similarly, the inward interlocking value between cities i and j is as follows:

$$C(j,i) = K_0 \sum_{m=1}^{n_2} K_1 X_m \tag{2}$$

where X is the number of branches of an enterprise in city i, and n2 is the total number of enterprises with headquarters in city j and branches in city i, and the total interlocking value of creative industries between cities i and j is defined as follows:

$$G(i,j) = C(i,j) + C(j,i) \tag{3}$$

Taking city i as the object, the interlocking direction of city j with i is as follows:

$$D(i,j) = [C(i,j) - C(j,i)]/[C(i,j) + C(j,i)] \tag{4}$$

To make the data comparable, the interlocking degree (0–100) can be obtained by normalization of all the interlocking values. The network strength of a city within the network is equal to the sum of its interlocking with all other cities, and the interlocking direction is calculated by dividing the difference between its outward and inward interlocking with all cities by the sum of the two. To study the influencing factors of regional development of Milan's creative industries, the Spearman correlation analysis was used to test the total, outward, and inward interlock of Milan and other cities with four categories of 20 indicators [49].

## 4. Discussion

### 4.1. Analysis of development characteristics of Milan's creative industries

(1) Firstly, a relatively simplified method is introduced in this research to analyze the overall pattern of the interlocking relationships between Milan's creative industries and European cities, and also between Milan's creative industries and major cities globally. Let the rough

**Table 3. Rough interlocking strength between Milan and other cities.**

| Level | Name of Cities and $C_i$ |
|---|---|
| Level 1 | Rome 86.0, **Paris 83.26**, **London 73.57** |
| Level 2 | **New York 69.74**, Madrid 68.72, Berlin 66.08, Munich 62.56, Barcelona 60.79, Vienna 60.79, |
| Level 3 | **Dubai 57.46**, Brussels 57.27, **Hong Kong 56.58**, Amsterdam 56.39, Frankfurt 55.96, Tokyo 54.82, Hambourg 54.63, Moscow 53.74, **Shanghai 53.51**, **Singapore 53.07**, Zurich 51.54, **Beijing 50.44** |
| Level 4 | Lisbon 49.78, Istanbul 49.34, Geneva 48.46, **Sydney 46.93**, Praha 45.37, Koln 44.49, Stuttgart 44.05, Lyon 43.17, Liverpool 43.17, Luxembourg 41.41, Copenhagen 40.97, Saint Petersburg 40.53, Warsaw 40.53 |
| Level 5 | Athens 39.65, Kiev 39.65, Antwerp 39.21, Stockholm 37.89, Nice 37.00, Dusseldorf 36.12, Basel 34.08, Budapest 34.36, Bilbao 33.92, Porto 33.92, Rotterdam 32.60, Marseille 32.60, Helsinki 31.28, Oslo 31.28 |
| Level 6 | The Hague 29.96, Seville 28.19, Lille 28.19, Eindhoven 27.75, Belgrade 25.99, Bremen 24.67, Tallinn 24.67, Zagreb 24.23, Riga 18.50, Minsk 17.62 |

interlocking strength of creative industries between Milan and City i be $C_i = M_i/N*100$, where N is the total number of multinational creative industry enterprises in Milan and $M_i$ is the number of the same multinational creative industry enterprises as Milan in City i. According to the calculation results, all cities were classified into 6 levels, as shown in Table 3: level 1 ($C_i \geq 70$), level 2 ($70 > C_i \geq 60$), level 3 ($60 > C_i \geq 50$), level 4 ($50 > C_i \geq 40$), level 5 ($40 > C_i \geq 30$), and level 6 ($C_i < 30$).

The creative industries' interlock between Milan and GaWC leading world cities is significant and greater than the average of the interlock with European cities, indicating that Milan's creative industries pay a lot of attention to the global level as well as the European regional level. Overall, the creative industry interlock intensity between Milan and most cities is at an intermediate level of 3–5, and all cities show a standard distribution trend at all levels. Besides Sydney, the other nine major cities of the world (including London and Paris) are above 50, which is at a higher level. After Rome, the first city to be associated with Milan is Paris (France) and the second is London (UK), and Milan is associated more closely with New York than with Asian cities. Milan's creative industries pay attention to the development at a global level reflects this city's strategy to become the world's creative center with extensive international influence [50]. It has a strong interlock with the leading world cities, but the overall gap is not wide, which also means that Milan's creative industries shows a certain balanced development trend at the world level.

Cities that are strongly associated with Milan's creative industries have obvious regional distribution differences, and thus formed an obvious creative industry cluster based on Milan. As shown in Figs 2 and 3, at the European level, Milan has a strong interlock with Central, Western, and Southern Europe (ranked in order), and a weak interlock with Northern and Eastern Europe. It is found that the interlocking strength levels between Milan and other cities in creative industries in Europe have obvious regional differentiation: Western Europe is evenly distributed at all levels, the cities in Central Europe are higher in overall classification, the cities in southern Europe are polarized, and the cities in Eastern Europe are generally weak. For cities in the Blue Banana region (also known as the European Megalopolis or the Liverpool—Milan Axis, is considered to be one of the populations, economy and creative hubs of Europe. The conceptualization of the region as a "Blue Banana" was developed in 1989 by RECLUS, a group of French geographers [51]), the mean and median of the interlock intensity with Milan are significantly higher than other European cities. Meanwhile, Paris, the city with the highest interlock intensity, is not a city in the Blue Banana region, which proves that Milan

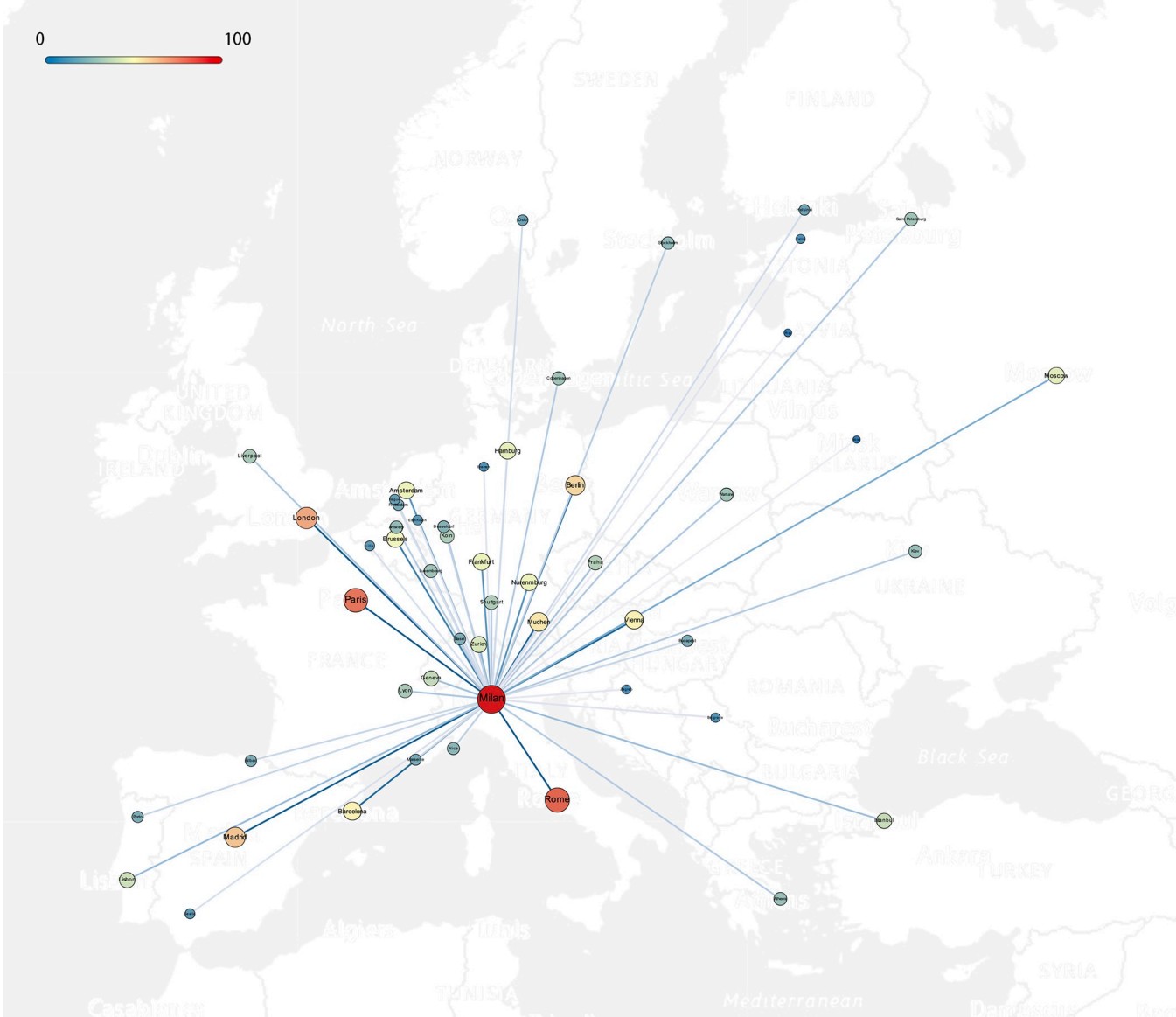

**Fig 2. Rough interlocking strength between Milan and major cities in Europe (Base map source: USGS National Map Viewer (public domain)).**

and cities in the Blue Banana region are more closely interlocked with creative industries, presenting an obvious creative industries cluster.

(2) Using the data of creative industries' enterprises, this study investigates the outward, inward, and interlock direction of Milan's creative industries and other cities, and then analyzes the characteristics of regional expansion and development of Milan's creative industries and the specific introduction of creative industries in other cities.

According to the analysis, at present, Milan is a typically extroverted city of creative industries, as seen in Fig 4, and the local creative industries have a strong radiating power. Specifically, it has a wide radiation range and effect on the creative industries of all 52 cities within the research scope. Further, it has strong radiation power, meaning that the export power of local creative industries is strong and significantly greater than the level of the imported

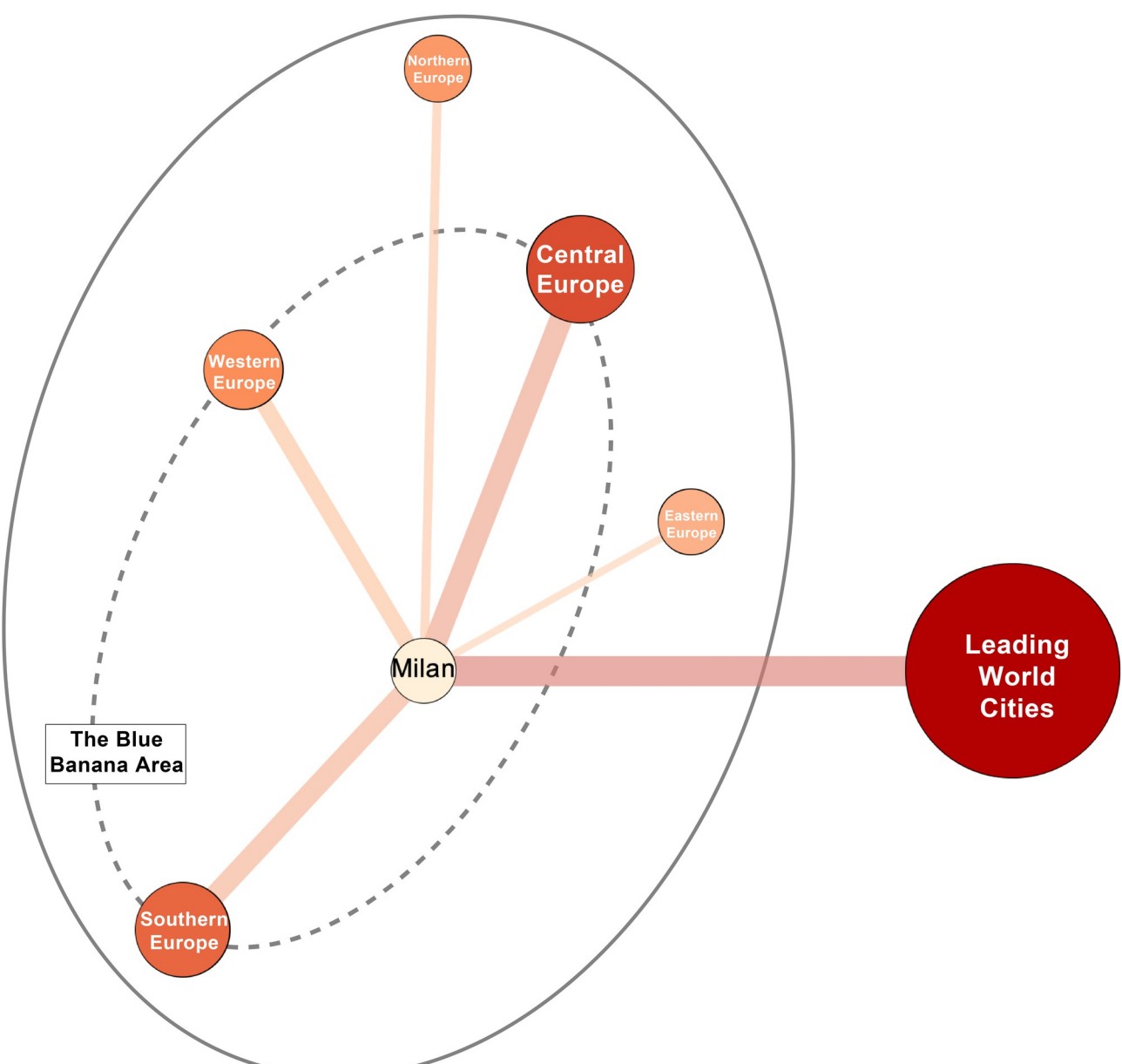

**Fig 3. Rough interlocking strength between Milan and major cities in different macro-regions in Europe and leading world cities.**

creative industries. The direction of interlock with all cities within the research scope is positive and in the range of 0 to 0.2 with most cities. Milan's emphasis on balanced expansion in Southern, Western, and Central Europe results in the average value of outward interlock of the three regions that are closely distributed, and the radiation of Northern and Eastern Europe are relatively weak.

From the evolution of the outward interlock, it can be seen that the external expansion of Milan's local creative industries focusses on key areas, and the expansion of scope and increase of interlocking intensity are almost conducted simultaneously in particular areas. These three regions also saw the greatest increase in outward interlock during the same period as the radiation spread to Southern, Central, and Western Europe. At present, the radiating effect of

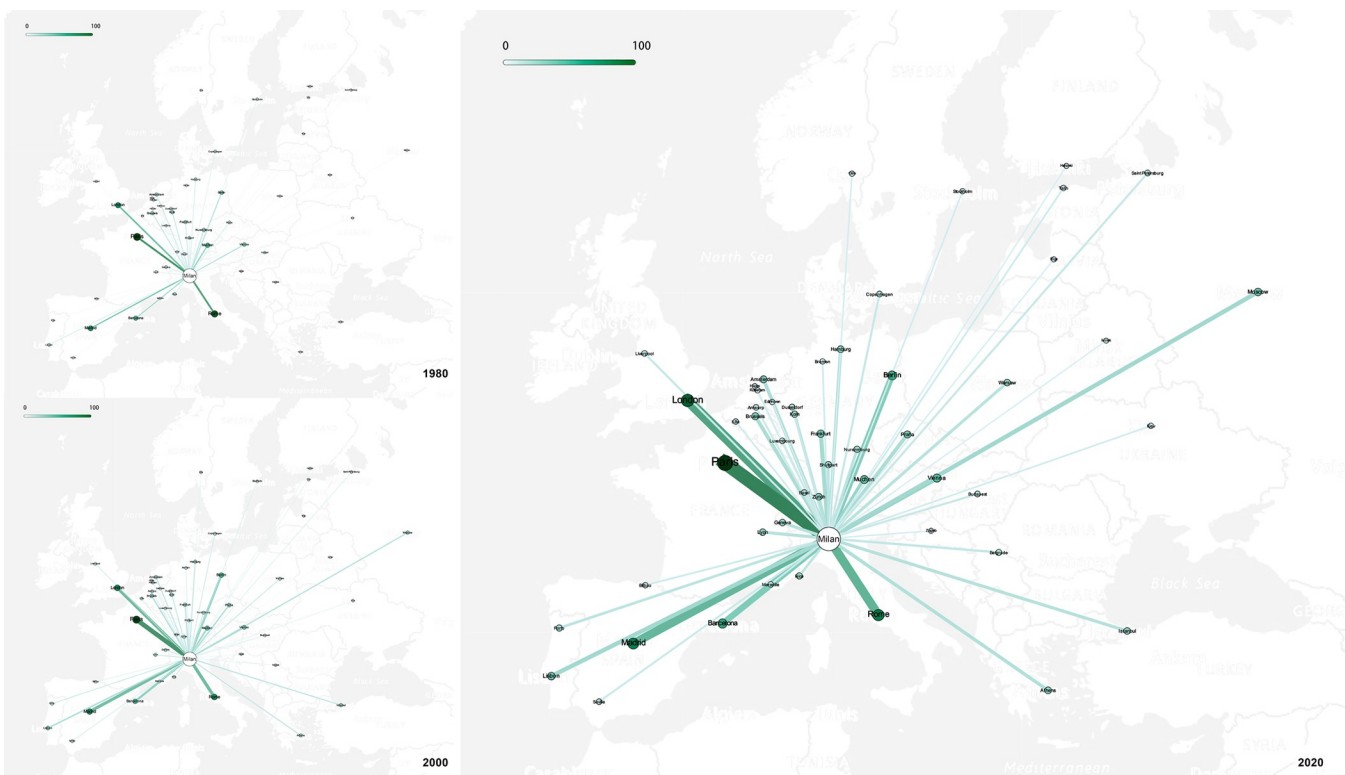

**Fig 4. Outward interlock between Milan and other cities.** (Base map source: USGS National Map Viewer (public domain)).

Milan on the creative industries of the cities within the research range can be divided into four levels: Paris, London, and Rome are significantly the strongest, and their extroverted interlock degree is greater than 60; Barcelona, Berlin, Vienna, and other cities, where the extroverted interlock degree is more than 30; the national capitals or important cities of Southern, Western, and Central Europe, such as Lisbon, Brussels, and Amsterdam, etc.; and the other cities as the fourth level.

Compared with the strong and extensive external radiation capacity of Milan's local creative industries, there are relatively few cities that have an influence on Milan's creative industries, which can be seen in Fig 5. The cities that have an influence on Milan's creative industries only account for about 65% of all cities within the research scope, most of which are distributed in Western, Central, and Northern Europe, and more than 60% of them are clustered in the "Blue Banana" area. Based on the combination of extroversion and introversion, the flow of creative industries' enterprises between Milan and Western and Central Europe, especially the cities in the "Blue Banana" region, is active; the cooperation is the closest between them, which also confirms the conclusion of the rough interlock analysis in Table 3; that is, Milan has formed an obvious regional creative industry cluster at the core.

(3) The data of creative industry enterprises are used to analyze the network structure and evolution characteristics of creative industries with Milan as the core.

As shown in Fig 6, the evolution of the interlock network of creative industries with Milan as its core can be roughly divided into four stages. Milan had not yet developed a strong network in the 1970s, with Paris and London being dominant. In the 1980s, the strength of Milan's network was enhanced with the development of Milan's creative industries. Cities in Central, Southern, and Western Europe, which were geographically close to Milan, began to

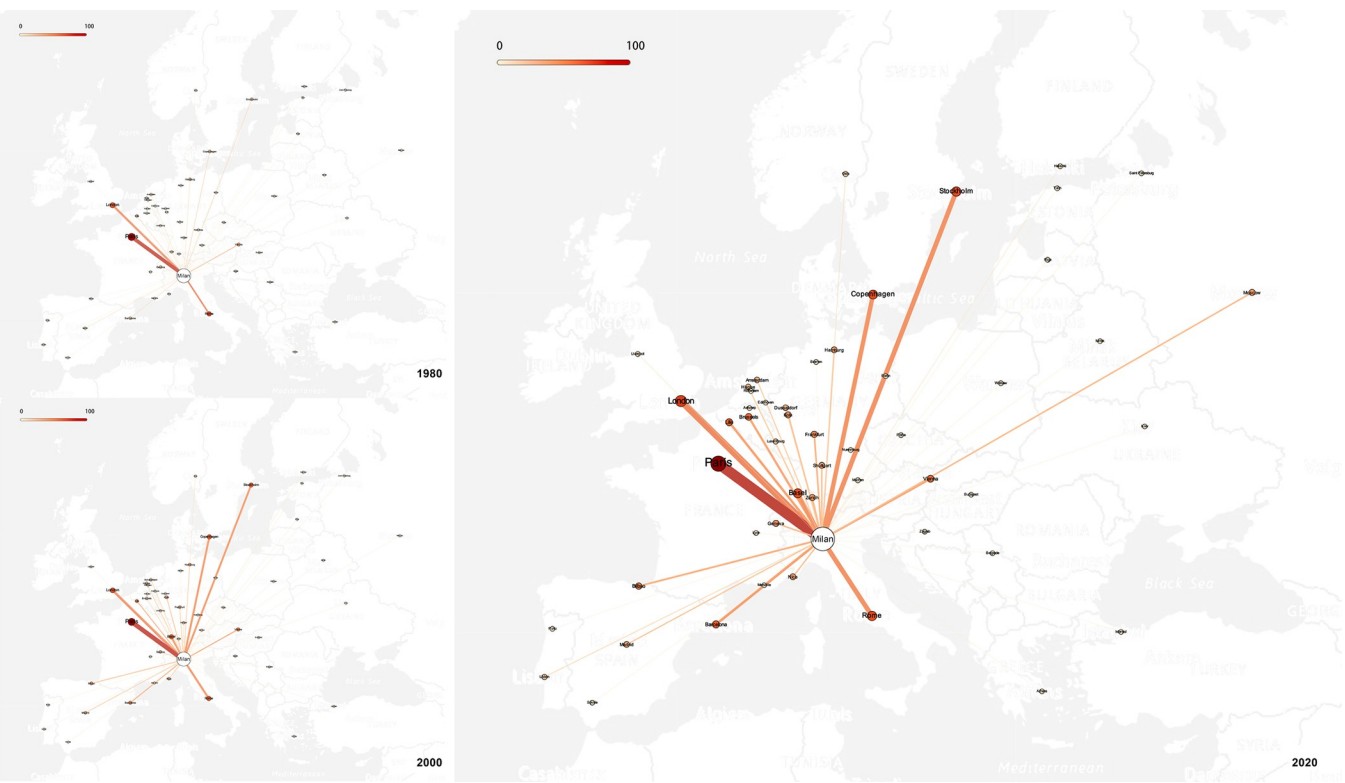

**Fig 5. Inward interlocking between Milan and other cities.** (Base map source: USGS National Map Viewer (public domain)).

join the network. Milan's creative industries boomed in the 1990s, its status in the network was strongly promoted, the "three cores" formed together with Paris and London, and its strongest triangular relationship between three cities also became the foundation of the network. Meanwhile, new strong links formed, such as between Milan and Madrid, and between Milan and Barcelona, so that the secondary core cities gradually became established. At the same time, the interlocking intensity of multiple levels began to differ more obviously, and the initial multi-dimensional structure of the network was formed. At this time, the coverage of the network was further expanded and most cities joined the network, except some cities in Eastern Europe. By the year 2000, the network had basically developed and matured, the status of secondary core cities had been enhanced, and the interlocks between different levels was further enriched. After 2000, the overall structure of the network became mature and stable, and no significant changes were made.

As shown in Fig 7, at present, the European creative industry network with Milan as its core presents a four-tier structure. Milan, Paris, and London are the first tier that established interlocks with all cities, and the triangle interlock formed by three cities is the strongest in the network. This is followed by Barcelona, Madrid, Copenhagen, Berlin, Vienna, and other cities as the secondary tier. These account for about 20% of all cities. They form the backbone of the network and are a strong interlock with the primary core, having an interlock with more than 70% of the cities within the scope of the research. Tier 3 cities are the main part of the network, accounting for around 50% and concentrated primarily in Central, Western, and Southern Europe. They are interlocked to 25% to 50% of the cities; among them are the core cities of the first and second level predominantly. Tier 4 cities are located mostly in Eastern Europe.

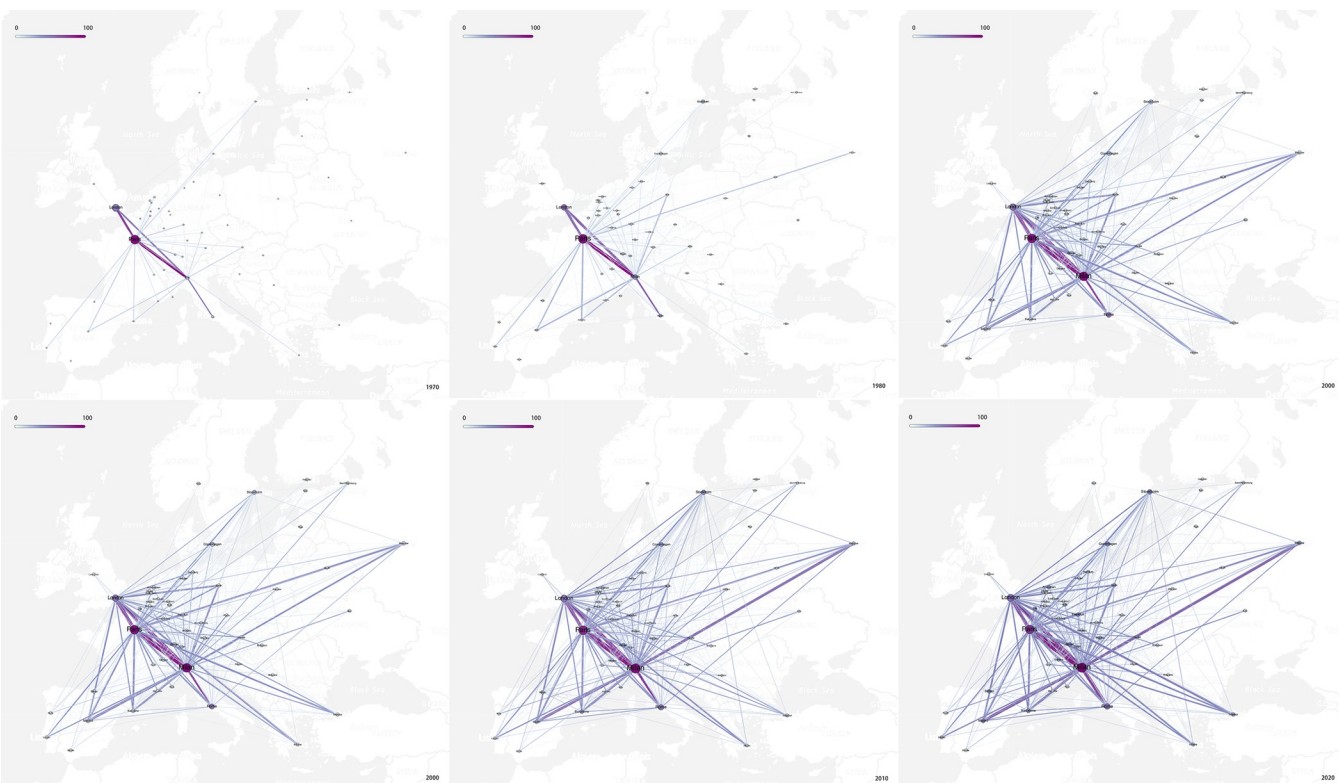

**Fig 6. Creative industry interlocking network based on Milan.** (Base map source: USGS National Map Viewer (public domain)).

With Milan as the core, the European creative industries network is diversified and multi-dimensional, as can be seen in Fig 8. There are many core and export-oriented creative cities (cities with a positive interlock direction relative to Milan in this study) in the network. It has broken away from the traditional structure of several dominants, forming Milan as a strong core and responding to the city's goal for the world's creative center. The wide range of network influence, phenomenon of many participating cities, and multi-dimensional structure all show that the flow of creative enterprises related to Milan has been active among European cities, providing a better environment for the regional development of Milan's creative industries. It is preliminarily concluded that the model of the European creative industries' network with Milan as its core is first built on a strong interlock between the absolute core, then expanded to a secondary core, gradually expanding its scope of influence, and finally strengthening the interlock at different levels.

## 4.2. Study on influencing factors of Milan's creative industries

Overall, the regional development of Milan's creative industries is gradually ridding itself of the limitation of geographic distance, which can be seen in Table 4. In the initial stage of development, the correlation between total interlocking, outward interlocking, and the distance factor were significantly negative; with the progress of time, the correlation first decreased to an almost insignificant value and then rises slightly. At the early stage of development, Milan's creative industries are obviously constrained by geographical distance. Currently, creative industries' enterprises tend to develop nearby and first establish interlocks with cities close to Milan, and then gradually rid themselves of location restrictions after the mature stage of

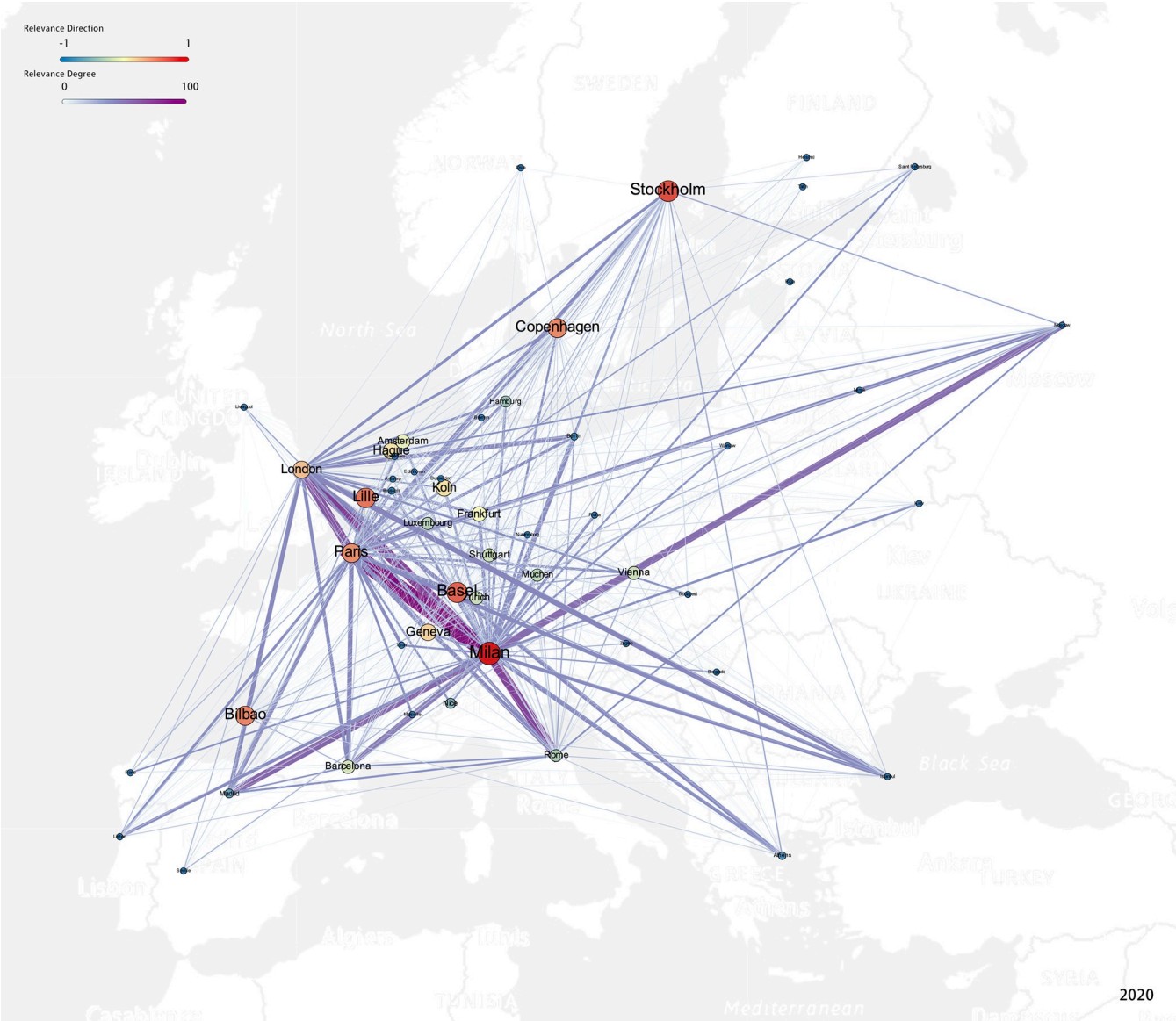

**Fig 7. Creative industries' interlocking network with the direction of interlocking.** (Base map source: USGS National Map Viewer (public domain)).

development. In the new development period after 2000, several new multinational enterprises in creative industries began to emerge in Milan, which are more inclined to expand to cities with close geographical distances at the initial stage of development.

As Table 5–8 show, the correlation between the four categories of factors and the three interlock types are ranked from strong to weak, as follows: technology, talent, infrastructure, and tolerance. Technology factors have the largest number of significantly related indicators, with the strongest correlation and no great change over time. It can be considered that technology factors have the strongest impact on the regional development of Milan's creative industries and have good stability. Talent factors have a strong correlation overall, and most indicators are significantly correlated with the three correlations to varying degrees, but have obvious changes over time. It is concluded that talent factors have a strong but unstable

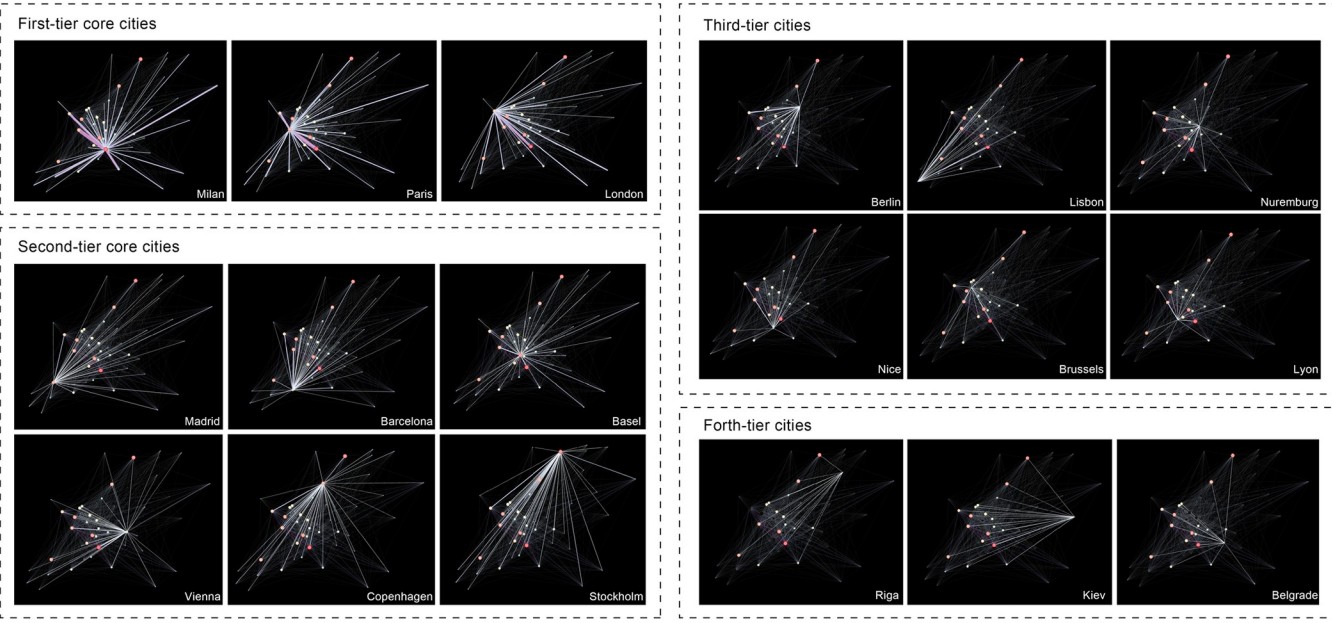

**Fig 8. The interlock scope of different tiers.**

influence on the regional development of Milan's creative industries. About half of the factors related to infrastructure are significantly correlated, with a strong correlation and little change over time, indicating that they have a certain impact on the regional development of Milan's creative industries and have good sustainability. Only a few factors of the tolerance category were significantly correlated with the three correlations, indicating that their influence was relatively minimal.

## 5. Conclusions

Currently, Milan has become a center of global creativity and design. This study uses a variety of data to analyze the development and evolution of its creative industries from a regional perspective, and to discusses its main development characteristics as one of the world's creative centers. As a result, this study contains conclusions and reflections on a quantitative basis, as follows:

**Table 4. The relationship between the distance to Milan and three interlock types.**

| Year | Total Interlock | Outward Interlock | Inward Interlock |
|------|-----------------|-------------------|------------------|
| 2020 | 0.028/-0.313* | 0.040/-0.294* | 0.049/-0.272 |
| 2010 | 0.043/-0/290* | 0.049/-0.283* | 0.065/-0.266 |
| 2000 | 0.088/-0.247 | 0.056/-0.275 | 0.072/-0.259 |
| 1990 | 0.026/-0.319* | 0.028/-0.314* | 0.092/-0.243 |
| 1980 | 0.002/-0.433** | 0.001/-0.443** | 0.501/-0.095 |
| 1970 | 0.000/-0.547** | 0.000/-0.542** | 0.118/-0.226 |

* Significance Sig./ Interlocking index

** means significant at 0.01 level

* means significant at 0.05 level

**Table 5. The relationship between technology indicators and three interlock types.**

| Indicators | | Total Interlock | Outward Interlock | Inward Interlock |
|---|---|---|---|---|
| R&D investment | 2010 | 0.010/0.374** | 0.052/0.285 | 0.002/0.439** |
| | 2018 | 0.005/0.391** | 0.034/0.304* | 0.003/0.421** |
| Community Design | 2010 | 0.000/0.581** | 0.000/0.554** | 0.001/0.460** |
| | 2016 | 0.000.0.513** | 0.01/0.475** | 0.001/0.470** |
| High-tech patents | 1990 | 0.000/0.587** | 0.000/0.556** | 0.000/0.508** |
| | 2000 | 0.000/0.507** | 0.002/0.444** | 0.001/0.481** |
| | 2010 | 0.000/0.513** | 0.001/0.456** | 0.000/0.502** |
| EU Trademark registrations | 2000 | 0.000/0.556** | 0.000/0.485** | 0.000/0.584** |
| | 2010 | 0.002/0.434** | 0.003/0.415** | 0.049/0.182* |
| Revenue of patents | 2010 | 0.060/0.271 | 0.189/0.194 | 0.001/0.384** |

\* Significance Sig./ Interlocking index

\*\* means significant at 0.01 level

\* means significant at 0.05 level

The regional development of Milan's creative industries contains many remarkable features. First, Milan pays attention to the creative industries' development on a global scale, reflects its macro development strategy. Compared with other European cities, Milan has established stronger and more balanced interlocks with leading world cities that is not limited to Europe. Strong international interlocks of creative industries are an important factor in Milan becoming a global creative center, responding to the city's ambition to become a world creative center, while creating its extensive international influence. Second, Milan is a typically extroverted and radiating city for creative industries, its foreign output is significantly greater than its imports in terms of creative industries. Compared to foreign creative industries, it pays more attention to the cultivation of local creative enterprises, resulting in local creative

**Table 6. The relationship between talent indicators and three interlock types.**

| Indicators | | Total Interlock | Outward Interlock | Inward Interlock |
|---|---|---|---|---|
| Educational level of residents aged 25–64 | 2000 | 0.023/0.315* | 0.116/0.223 | 0.000/0.493** |
| | 2010 | 0.043/0.285* | 0.197/0.184 | 0.000/0.472** |
| | 2019 | 0.01/0.357* | 0.059/0.266 | 0.001/0.470** |
| Students in higher education | 2010 | 0.000/0.541** | 0.000/0.570** | 0.024/0.323* |
| | 2017 | 0.000/0.567** | 0.000/0.564** | 0.006/0.386** |
| High-tech employment | 2010 | 0.001/0.478** | 0.010/0.373** | 0.000/0.492** |
| | 2019 | 0.006/0.388** | 0.020/0.331* | 0.019/0.335* |
| HRST | 2000 | 0.033/0.311* | 0.220/0.138 | 0.002/0.450** |
| | 2010 | 0.086/0.253 | 0.392/0.128 | 0.001/0.456** |
| | 2019 | 0.086/0.243 | 0.295/0.149 | 0.005/0.390** |
| R&D personnel status | 2000 | 0.349/0.14 | 0.551/0.089 | 0.119/0.231 |
| | 2010 | 0.017/0.346* | 0.047/0.307* | 0.002/0.434** |
| | 2019 | 0.001/0.464** | 0.003/0.417** | 0.010/0.363* |
| Employment in culture, creative, entertainment | 2017 | 0.886/0.021 | 0.727/0.050 | 0.426/-0.114 |

\* Significance Sig./ Interlocking index

\*\* means significant at 0.01 level

\* means significant at 0.05 level

**Table 7. The relationship between infrastructure indicators and three interlock types.**

| Indicators | | Total Interlock | Outward Interlock | Inward Interlock |
|---|---|---|---|---|
| Percentage of households with broadband access | 2010 | 0.366/0.132 | 0.881/0.022 | 0.001/0.468** |
| | 2019 | 0.370/0.128 | 0.755/0.045 | 0.017/0.332* |
| Highway construction | 2010 | 0.125/0.222 | 0.137/0.215 | 0.126/0.222 |
| | 2017 | 0.404/0.122 | 0.385/0.127 | 0.283/0.156 |
| Aviation condition | 2010 | 0.000/0.655** | 0.000/0.594** | 0.000/0.505** |
| | 2019 | 0.000/0.667** | 0.000/0.628** | 0.000/0.556** |
| Railway construction | 2010 | 0.010/0.365** | 0.003/0.411** | 0.178/0.195 |
| | 2018 | 0.012/0.357* | 0.005/0.395** | 0.183/0.193 |
| Number of facilities for visitors | 2010 | 0.000/0.513** | 0.000/0.466** | 0.003/0.413** |
| | 2017 | 0.000/0.534** | 0.000/0.499** | 0.001/0.473** |

* Significance Sig./ Interlocking index

** means significant at 0.01 level

* means significant at 0.05 level

industries with strong development and rapid expansion regionally and globally. Milan is embodied in strong radiation ability and has radiation effects on 52 cities, so its interlock directions are all positive. Third, the regional development of Milan's creative industries is gradually ridding themselves of the constraints of geographical distance, which is not limited to establishing strong links with nearby cities. This gradually expands the development scope of its creative industries globally. Fourth, it can be seen from the evolution of the outward interlock of Milan's creative industries that the external expansion of Milan's local creative industries at the regional level focuses on key areas rather than developing evenly.

Milan's creative industry development also relies on strong regional support to form an innovation cluster. First, cities with strong two-way links with Milan's creative industries tend to gather in certain regions, forming regional creative industry clusters cooperating with and relying on Milan. Secondly, through the four stages of evolution, the interlocking network of creative industries with Milan at its core has gradually built a multi-dimensional and four-level structure consisting of three cores. Within the network, there are many core cities with strong strengths, export-oriented creative cities, and strong core forces, forming strong regional support for the development of Milan. The large number of participating cities and their wide

**Table 8. The relationship between tolerance indicators and three interlock types.**

| Indicators | | Total Interlock | Outward Interlock | Inward Interlock |
|---|---|---|---|---|
| Foreign-born population | 2010 | 0.000/0.527** | 0.001/0.475** | 0.002/0.439** |
| | 2017 | 0.000/0.519** | 0.001/0.464** | 0.004/0.408** |
| Gender employment gap | 2010 | 1.000/0.000 | 0.791/0.039 | 0.446/-0.107 |
| | 2019 | 0.511/0.096 | 0.254/0.166 | 0.277/-0.158 |
| Proportion of women in higher education | 2010 | 0.039/0.291* | 0.042/0.246* | 0.018/0.329* |
| | 2018 | 0.049/0.234* | 0.037/0.268* | 0.009/0.362** |
| Percentage of economically active women | 2010 | 0.045/-0.260* | 0.023/-0.219* | 0.006/-0.378* |
| | 2018 | 0.032/-0.214* | 0.028/-0.209* | 0.047/-0.269* |

* Significance Sig./ Interlocking index

** means significant at 0.01 level

* means significant at 0.05 level

range of influence also show the active flow of European creative industries' enterprises, providing a better environment for the cooperation and development of European creative industries.

The regional development of Milan's creative industries has significant influencing factors. In general, the order of influence from strong to weak is technology, talent, infrastructure, and tolerance. Milan is most inclined to establish two-way relations with cities with a high investment in R&D and many scientific and technological patents. It is more inclined to establish cooperation with cities with high education levels and many scientific and technological talents. To a certain extent, it is also affected by infrastructure factors, such as the aviation and railway system, and tolerance factors such as the foreign population.

Today, to become a global center of creativity and design, it is necessary to pay attention to the establishment of strong interlocks with important cities globally and enhance one's global influence, paying attention to the cultivation and external expansion of local creative industries, to the establishment of cooperation with cities with a strong scientific and technological talent base, and through building innovation cooperation communities with cities in the region. This gradually builds a multi-dimensional, multi-center, regional creative network to support urban development. Based on the data, this study uses Milan's creative industries' development as an example, which is of great significance to cities in developing countries that want to become international creative centers.

## Acknowledgments

Thank you to teachers and colleagues in Politecnico di Milano, Italy, and Tongji University, China for providing the feasibility for this research.

## Author Contributions

**Conceptualization:** Cheng Shen, Xinyi Zhang.

**Data curation:** Xiang Li.

**Formal analysis:** Xiang Li.

**Investigation:** Cheng Shen.

**Methodology:** Cheng Shen, Xiang Li.

**Resources:** Cheng Shen.

**Software:** Xinyi Zhang.

**Validation:** Xiang Li.

**Visualization:** Xinyi Zhang.

**Writing – original draft:** Xiang Li.

**Writing – review & editing:** Cheng Shen, Xiang Li.

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
