## [Decision Letter · Decision Letter 0]

6 Dec 2022

PONE-D-22-22397The development of global creative city centers from a regional perspective: A case study of Milan’s creative industriesPLOS ONE

Dear Dr. Li,

Thank you for submitting your manuscript to PLOS ONE. After careful consideration, we feel that it has merit but does not fully meet PLOS ONE’s publication criteria as it currently stands. Therefore, we invite you to submit a revised version of the manuscript that addresses the points raised during the review process.

We look forward to receiving your revised manuscript.

Kind regards,

Xianzhong Cao

Academic Editor

PLOS ONE

Journal Requirements:

NO authors have competing interests

3. We note that you have stated that you will provide repository information for your data at acceptance. Should your manuscript be accepted for publication, we will hold it until you provide the relevant accession numbers or DOIs necessary to access your data. If you wish to make changes to your Data Availability statement, please describe these changes in your cover letter and we will update your Data Availability statement to reflect the information you provide

5. We note that Figures 2, 3, 4, 5 and 6 in your submission contain [map/satellite] images which may be copyrighted. All PLOS content is published under the Creative Commons Attribution License (CC BY 4.0), which means that the manuscript, images, and Supporting Information files will be freely available online, and any third party is permitted to access, download, copy, distribute, and use these materials in any way, even commercially, with proper attribution. For these reasons, we cannot publish previously copyrighted maps or satellite images created using proprietary data, such as Google software (Google Maps, Street View, and Earth). For more information, see our copyright guidelines: http://journals.plos.org/plosone/s/licenses-and-copyright.

a. You may seek permission from the original copyright holder of Figures 2, 3, 4, 5 and 6 to publish the content specifically under the CC BY 4.0 license.  

Reviewers' comments:

Reviewer's Responses to Questions

**Comments to the Author**

1. Is the manuscript technically sound, and do the data support the conclusions?

Reviewer #1: Partly

2. Has the statistical analysis been performed appropriately and rigorously? 

Reviewer #1: I Don't Know

3. Have the authors made all data underlying the findings in their manuscript fully available?

Reviewer #1: No

4. Is the manuscript presented in an intelligible fashion and written in standard English?

Reviewer #1: Yes

5. Review Comments to the Author

Reviewer #1: Review on the paper entitled “The development of global creative city centers from a regional perspective: A case study of Milan’s creative industries”.

In this paper, the authors attempt to demonstrate the development of Milan’s creative industry and how the city is interconnected with some selected cities and what position it occupies in the network. In the post-industrial era characterizing developed countries worldwide, cities have tried to position themselves as global, national, and regional financial centers, homes to advanced producer service firms, and sites of creative industries. Some cities have become quite successful in establishing themselves as central cities of such activities. Milan serves a good example of being a center of creative industries. So, the research the authors conducted is up to date, and may attract the attention of the international scientific community. However, I have many concerns regarding the manuscript that the authors must carefully address to make the paper suitable for publication. In the followings, I am providing a detailed list for the authors about my observations:

1) First of all, I could not recognize the main goals of the research as they were not explicitly demonstrated. In other words, I miss the research questions from the manuscript.

2) The authors mention several times in the manuscript that it is Milan’s strategy to become “the world’s creative center” (see, for example, lines 222-223). Furthermore, they also assert that the creative industries have certain strategies (see, for example, lines 224-225). However, I found no reference about such strategies. So, where do these pieces of information come from? Does Milan really have a strategy to become “the world’s creative center”? How should the readers picture the strategy of the creative industries?

3) I think the title of the manuscript is not appropriately chosen. For me, the term “city center” refers to the inner urban area of a city rather than city’s position in a network. This term also pops up in the text. Please, check it.

4) The keywords should be rewritten. Because of their too general meaning, the following keywords would not increase the visibility of the paper: region, development characteristics, and influencing factors.

5) When I was first reading that the authors wanted to investigate Milan’s creative industry from a regional perspective, I thought that the geographical scope of the research would be Greater Milan or Lombardy. Then it turned out that the authors considered Europe as the region and also involved some world city as defined by GaWC in the analysis. Please, reconsider the use of the term “regional” in this context, because in Europe, it refers to a spatial unit under the national level.

6) This sentence should be rewritten (lines 92-94): “Thereafter, Hall and other scholars did not deny that the characteristics of flow, which began to appear in Western Europe.”

7) In line 95, this can be read: ‘…the research method based on the concept of "space of flow"…’ I am wondering what this research method could be (i.e., what is based on the concept of space of flow).

8) In line 119, the authors write that “the scope of this research is selected”, but it is not clear why and how the scope of the research is selected. Please, reconsider this argument.

9) In lines 119-122, the authors put forth that “According to the European Creative City Ranking [34], List of European Cities and Metropolitan Areas based on Population [35], and Ranking of European Metropolitan Areas based on GDP [36], 52 European cities were identified as the scope of this research.” I think the cities involved in the analysis were chosen by the authors rather than the European Creative City Ranking, etc. Please, rewrite the sentence.

10) In Table 1, the authors classified cities into different macro-regions of Europe, but it was not referred in the manuscript that on what basis they created the classification. Is there an international organization who use this classification or the authors arbitrary created those city classes?

11) The official name of the city “Hague” is “The Hague”. Please, check it.

12) In lines 136-148, the authors briefly describe the evolution of Milan’s fashion design industry as an example of creative industries. However, it is not clear which industries are considered as creative industries, especially in the case of Milan. We have only been informed that the fashion design industry belongs to the creative industries.

13) In line 144, the authors write that ‘…creative industries in Milan gradually expanded from the original "quadrilatero della moda" to the entire city and formed a cluster effect.’ I think it is not possible to form an “effect”. Please check the meaning of this expression.

14) In line 162, the authors put forth that they classified enterprises whether they were multinational or not. I am wondering where the data came from and how they found out which company were multinational.

15) As for the NUTS classification, I assume that some NUTS 3 regions cover a larger area than the city, the authors chose to analyze.

16) The authors involved many indicators in the study to investigate the 52 cities (see, Table 2). They picked “net migration rate” as an indicator and I assume that they considered this indicator positive if the ratio is high. I just want to add here, that this indicator does not necessarily reflect the tolerance of the local society, primarily not in Europe. Italy has witnessed the influx of migrant people recently, and Italian people have gradually lost tolerant towards migrants which has eventually resulted in the support of a populist anti-migration right-wing party who now governs Italy.

17) I think the number of registered vehicles has nothing to do with the cities’ infrastructure. It rather shows that people can afford to buy cars and might reflect the underdevelopment of the public transportation system.

18) In lines 215-216, the authors conclude that “Milan's creative industries pay more attention to the global level rather than the European regional level.” Based on the Table 3, this argument is not quite right. First, the authors selected some Alpha world cities from the GaWC list that construct strong links with cities across the word. This is the reason why they are marked as Alpha world cities. Second, as far as I see, there are more European cities among Milan’s most important connections than non-European world cities.

19) In line 235 (and several times later), the authors mention the term “Blue Banana” in the manuscript referring to a region of highly developed cities lying between England and Northern Italy. Please add a short description of the Blue Banana region to the manuscript. In addition, I would like to add that the term “Blue Banana” has been developed by a French journalist and it lacks scientific validation. The European Commission refers to Europe’s core region as the pentagon, one of which center is Milan.

20) Table 3 shows a six-level classification of cities, whereas the authors talk about a four-level classification several times in the manuscript (for example, in lines 265 and 303). Please, check it.

21) In line 315, the authors mention the term “export-oriented creative city”. I am not sure how an export-oriented creative city could be defined. Please provide a definition of this term.

22) In line 338, the authors write that there is a “correlation between Milan and research cities’ creative industries total interlocking…” Could you please specify what a “research city” is?

23) Data acquisition might have been a very interesting part of the paper (see, Figure 1); however, unfortunately, data are not included in the manuscript even though the authors asserted it.

24) As for Figure 2-2, I recommend the authors to change the expression “Main Cities in the World” to “Leading World Cities” or something like this.

6. PLOS authors have the option to publish the peer review history of their article (what does this mean?). If published, this will include your full peer review and any attached files.

Reviewer #1: No

---

## [Author Response · Author response to Decision Letter 0]

9 Jan 2023

Dear reviewer:

We are very grateful to your careful comments for the manuscript and we have learned a lot from your rich knowledge and rigor in reviewing manuscripts. According with your advice, we tried our best to amend the relevant parts and made some changes in the manuscript. All of your comments were answered below. And we list the changes and marked in revised paper.

We appreciate for Reviewer’s warm work earnestly, and hope that the correction will meet with approval. Should you have any questions, please contact us without hesitate. 

Once again, thank you very much for your comments and suggestions.

Yours Sincerely, 

The authors

Comments and Suggestions:

Comment 1. First of all, I could not recognize the main goals of the research as they were not explicitly demonstrated. In other words, I miss the research questions from the manuscript.

Response 1: We are very sorry for not explicitly demonstrating the goal of the research. We have added the research goal in the third section: Materials and methods (lines 146-153).

The goal of this study is to analyze the development of creative industries in Milan, a global creative center, by using multi-source data within the scope of a Europe-wide region. The study takes the data of creative industry enterprises as relational data, analyzes the interlock relationship of the creative industry built by Milan and cities in the region, the characteristics and development trends of the interlock network with Milan as the core and combined with a number of indicators, to further explore the influencing factors of regional development of Milan's creative industry. Through the study of a global creative center from a regional perspective, references for emerging creative cities and those that require industrial transformation can be achieved.

Comment 2. The authors mention several times in the manuscript that it is Milan’s strategy to become “the world’s creative center” (see, for example, lines 222-223). Furthermore, they also assert that the creative industries have certain strategies (see, for example, lines 224-225). However, I found no reference about such strategies. So, where do these pieces of information come from? Does Milan really have a strategy to become “the world’s creative center”? How should the readers picture the strategy of the creative industries?

Response 2: We are very sorry for not including the references, and we have made the following modifications:

1. Milan does, in fact, have a strategy to become a global city and a center of creativity and innovation. We have added references to the manuscript (line 336). As mentioned in the PGT Milano 2030 planning document, Milan plans to become a global city, creating physical and non-physical interlocks with a larger area and attracting a wider market; In the fields of fashion and design, media and communication and so on, it will strengthen the interlock with advanced service networks, strengthen cooperation between enterprises, and become a global creative and innovation capital.

2. The analysis in this study may express that the development of Milan's creative industry responding to the strategy of Milan to become a global creative center. The strategy of Milan’s creative industry mentioned in this study may be an inference and we have revised this part: keeping the expression for the strategy of Milan to become a global creative center but remove the expression for the strategy of creative industries in Milan (lines 335-338).

Comment 3. I think the title of the manuscript is not appropriately chosen. For me, the term “city center” refers to the inner urban area of a city rather than city’s position in a network. This term also pops up in the text. Please, check it.

Response 3: We are sorry for using the inappropriate expression. We have made correction in the title and relevant parts of the manuscript (line 95).

Comment 4. The keywords should be rewritten. Because of their too general meaning, the following keywords would not increase the visibility of the paper: region, development characteristics, and influencing factors.

Response 4: Thanks, and we have rewritten the keyworks as the Reviewer suggested. The keywords now are: Creative industries; Milan; global creative centers; development trends; creative network; influencing factors.

Comment 5. When I was first reading that the authors wanted to investigate Milan’s creative industry from a regional perspective, I thought that the geographical scope of the research would be Greater Milan or Lombardy. Then it turned out that the authors considered Europe as the region and also involved some world city as defined by GaWC in the analysis. Please, reconsider the use of the term “regional” in this context, because in Europe, it refers to a spatial unit under the national level.

Response 5: Thanks, and it’s really true as the Reviewer suggested. To solve this problem, we made the following modifications: in the introduction section where the term “region” is mentioned for the first time (lines 92-94), we have added explanations to emphasize that in this study this concept is not refer to a specific spatial unit in the European context, but the broad concept of a large area with unclear boundaries.

Comment 6. This sentence should be rewritten (lines 92-94): “Thereafter, Hall and other scholars did not deny that the characteristics of flow, which began to appear in Western Europe.”

Response 6: Thanks, and we have rewritten the sentence according to the Reviewer’s comments (lines 134-136). Thereafter, Hall and some other scholars hold that the characteristics of flows began to appear in Western Europe.

Comment 7. In line 95, this can be read: ‘...the research method based on the concept of "space of flow"...’ I am wondering what this research method could be (i.e., what is based on the concept of space of flow).

Response 7: We are sorry for the confusions here and we have made some correction and added some explanations in this section (lines 138-140). 

Under the concept of space of flow, "flow" reflects the interaction between cities as real relational data. This prompts the research of regional spatial structure to shift from the internal characteristics of the city to the external relations of the city, and prompts the focus to the urban network and interlock relations. Subsequent studies based on various flows (mentioned in the literature review section, lines 141-170) can be considered to have originated under this concept. Also, there may be some differences in specific analysis methods due to different types of the data.

Comment 8. In line 119, the authors write that “the scope of this research is selected”, but it is not clear why and how the scope of the research is selected. Please, reconsider this argument.

Response 8: We are sorry for the unclear expression and we have added some content in this section about why and how the scope of the research is selected (lines 183-197). 

“In order to cover the whole Europe, make the selected cities representative, and facilitate data acquisition and analysis, taking comprehensive factors such as innovation capacity, economy, urban size, population, location, and so on into consideration and referring some lists of European cities such as the European Creative City Ranking, List of European Cities and Metropolitan Areas based on Population, Ranking of European Metropolitan Areas based on GDP etc., we selected 52 European cities as the scope of this research.” With such a choice, this study can achieve the research goal set by the author and explore the development of creative industries of Milan from a larger regional scale.

Comment 9. In lines 119-122, the authors put forth that “According to the European Creative City Ranking [34], List of European Cities and Metropolitan Areas based on Population [35], and Ranking of European Metropolitan Areas based on GDP [36], 52 European cities were identified as the scope of this research.” I think the cities involved in the analysis were chosen by the authors rather than the European Creative City Ranking, etc. Please, rewrite the sentence.

Response 9: It is true and we have rewritten the sentence according to the Reviewer’s comments (lines 183-197). 

Comment 10. In Table 1, the authors classified cities into different macro-regions of Europe, but it was not referred in the manuscript that on what basis they created the classification. Is there an international organization who use this classification or the authors arbitrary created those city classes?

Response 10: We are sorry that we didn't include the reference. We have made supplementary explanations and added references above Table 1 (lines 198-201). We refer to the geographic location division of European countries in The World Factbook (also known as the CIA World Factbook), compiled by the Central Intelligence Agency. We also mentioned that the division of marco-regions here is only used for the statistical convenience and does not imply any assumption regarding political or other affiliation of countries or territories.

Comment 11. The official name of the city “Hague” is “The Hague”. Please, check it.

Response 11: We are very sorry for our incorrect writing. We have corrected it.

Comment 12. In lines 136-148, the authors briefly describe the evolution of Milan’s fashion design industry as an example of creative industries. However, it is not clear which industries are considered as creative industries, especially in the case of Milan. We have only been informed that the fashion design industry belongs to the creative industries.

Response 12: We are sorry for the confusions. In fact, there is some explanation on the definition and classification of creative industry in the section of core concept definitions, but the explanation may not be clear. Therefore, we added content in this chapter (lines 227-228), emphasizing that the classification and data acquisition of creative industries of Milan in this research followed the nine categories of creative industries determined by DCMS.

Comment 13. In line 144, the authors write that ‘...creative industries in Milan gradually expanded from the original "quadrilatero della moda" to the entire city and formed a cluster effect.’ I think it is not possible to form an “effect”. Please check the meaning of this expression.

Response 13: We are very sorry for the inappropriate expression and we have made correction to remove this expression (line 236).

Comment 14. In line 162, the authors put forth that they classified enterprises whether they were multinational or not. I am wondering where the data came from and how they found out which company were multinational.

Response 14: Thanks, and as explained in the section of data acquisition, we first obtained the POI data of the creative industry in Milan, because there is no statistics on whether it is a multinational enterprise or not in the POI data, so we adopted the manual classification method, searched the information of the enterprise according to the name, and identified its headquarters registration place and whether it has branches in other countries or cities. Finally, we can label whether it is a multinational enterprise or not.

Comment 15. As for the NUTS classification, I assume that some NUTS 3 regions cover a larger area than the city, the authors chose to analyze.

Response 15: Thanks, and it is true as the Reviewer said. We have made some explanations in the section of data acquisition (line 278-279), due to the availability of the data, we cannot directly obtain all the indicator data with the city as the statistical unit on Eurostat. Therefore, some data uses NUTS as the statistical unit to replace the data of the core city in the NUTS. At the same time, in the correlation analysis section based on these indicators, a correlation analysis method based on data ranking is adopted, and the same index uses the same standard statistical unit, so we think that the analysis results also have certain reference value.

Comment 16. The authors involved many indicators in the study to investigate the 52 cities (see, Table 2). They picked “net migration rate” as an indicator and I assume that they considered this indicator positive if the ratio is high. I just want to add here, that this indicator does not necessarily reflect the tolerance of the local society, primarily not in Europe. Italy has witnessed the influx of migrant people recently, and Italian people have gradually lost tolerant towards migrants which has eventually resulted in the support of a populist anti-migration right-wing party who now governs Italy.

Response 16: Thanks for advice and guidance. We have removed the “net migration rate” from the tolerance indicators (Table 2 and Table 8) and made some modifications accordingly in the analysis part. The authors once studied in Italy and is familiar with the situation you mentioned. Under your guidance, we have carefully studied relevant policies and literature, and this indicator does not necessarily reflect the tolerance of the local society, which is our previous misunderstanding. So, we decided to remove that indicator.

Comment 17. I think the number of registered vehicles has nothing to do with the cities’ infrastructure. It rather shows that people can afford to buy cars and might reflect the underdevelopment of the public transportation system.

Response 17: It is really true as Reviewer suggested, and we have removed the “number of registered vehicles” from the infrastructure indicators (Table 2 and Table 7) and made some modifications accordingly in the analysis part.

Comment 18. In lines 215-216, the authors conclude that “Milan's creative industries pay more attention to the global level rather than the European regional level.” Based on the Table 3, this argument is not quite right. First, the authors selected some Alpha world cities from the GaWC list that construct strong links with cities across the word. This is the reason why they are marked as Alpha world cities. Second, as far as I see, there are more European cities among Milan’s most important connections than non-European world cities.

Response 18: It is really true as the Reviewer suggested and the argument here is not accurate. We have made some changes here to avoid comparisons to make the argument rigorous (lines 322-338). 

The creative industries’ interlock between Milan and GaWC leading world cities is significant and greater than the average of the interlock with European cities, indicating that Milan's creative industries pay a lot of attention to the global level as well as the European regional level.

Comment 19. In line 235 (and several times later), the authors mention the term “Blue Banana” in the manuscript referring to a region of highly developed cities lying between England and Northern Italy. Please add a short description of the Blue Banana region to the manuscript. In addition, I would like to add that the term “Blue Banana” has been developed by a French journalist and it lacks scientific validation. The European Commission refers to Europe’s core region as the pentagon, one of which center is Milan.

Response 19: Thanks a lot for advice and guidance. We have added a short description of the “Blue Banana” and the references in this section where the term is first mentioned (lines 349-351). 

Comment 20. Table 3 shows a six-level classification of cities, whereas the authors talk about a four-level classification several times in the manuscript (for example, in lines 265 and 303). Please, check it.

Response 20: We are very sorry for the confusions and we have made correction to distinguish the two parts. In the section of network analysis (lines 428-436), “a four-tier structure” is used to express the classification. In fact, these two classifications are classified in different situations and different calculation process, and they do not conflict. The former is a classification based on the calculated value under the rough interlock degree calculation method. The latter is a classification of a city in the creative network according to its status which may be a more general description.

Comment 21. In line 315, the authors mention the term “export-oriented creative city”. I am not sure how an export-oriented creative city could be defined. Please provide a definition of this term.

Response 21: We are sorry that we did not explain its definition clearly in manuscript and we have added a short description in this section (lines 440-441). According to the interlock direction calculated by Formula 4 and the result shown in Figure 7, in the creative industry interlock network with Milan as the core, the cities with positive interlock direction (which means the outward interlock of local creative industries is greater than the inward interlock brought by foreign creative industry) can be regarded as "export-oriented creative city" in the context of this study. The export-oriented creative city defined in this study is based on the export-oriented economy, because creativity, like economy, is a factor of flows. In addition, the concept here is relative to Milan. In the context of this study, this city is an "export-oriented creative city" relative to Milan, but it does not represent the general situation of this city.

Comment 22. In line 338, the authors write that there is a “correlation between Milan and research cities’ creative industries total interlocking...” Could you please specify what a “research city” is?

Response 22: We are very sorry for using the wrong expression. In fact, what we originally intended to express was the interlock between Milan and 52 cities within the research scope so “research cities” are the cities within the research scope. We have made correction for this part and removing this expression (lines 464-465).

Comment 23. Data acquisition might have been a very interesting part of the paper (see, Figure 1); however, unfortunately, data are not included in the manuscript even though the authors asserted it.

Response 23: We are sorry for that. According to the PLOS Data policy, we will provide data repository information for our data at acceptance. We are working on it and we have deposited it to a public repository on github:

https://github.com/ScaithyShen/Creative-industries-of-Milan.git

Comment 24. As for Figure 2-2, I recommend the authors to change the expression “Main Cities in the World” to “Leading World Cities” or something like this.

Response 24: Thanks, and we have made correction according to your suggestion (in Figure 2-2, line 322, 337 and 504).

We Special thanks to you for your good comments.

Again, thank you for giving us the opportunity to strengthen our manuscript with your valuable comments and queries. We have worked hard to incorporate your feedback and hope that these revisions persuade you to accept our submission.

---

## [Editor Report · Decision Letter 1]

5 Feb 2023

The development of global creative centers from a regional perspective: A case study of Milan’s creative industries

PONE-D-22-22397R1

Dear Dr. Li,

We’re pleased to inform you that your manuscript has been judged scientifically suitable for publication and will be formally accepted for publication once it meets all outstanding technical requirements.

Kind regards,

Xianzhong Cao

Academic Editor

PLOS ONE
---

## [Editor Report · Acceptance letter]

30 Mar 2023

PONE-D-22-22397R1 

The development of global creative centers from a regional perspective: A case study of Milan’s creative industries 

Dear Dr. Li:

I'm pleased to inform you that your manuscript has been deemed suitable for publication in PLOS ONE. Congratulations! Your manuscript is now with our production department. 

Kind regards, 

on behalf of

Professor Xianzhong Cao 

Academic Editor

PLOS ONE